# The MYB-CC Transcription Factor PHOSPHATE STARVATION RESPONSE-LIKE 7 (PHL7) Functions in Phosphate Homeostasis and Affects Salt Stress Tolerance in Rice

**DOI:** 10.3390/plants13050637

**Published:** 2024-02-26

**Authors:** Won Tae Yang, Ki Deuk Bae, Seon-Woo Lee, Ki Hong Jung, Sunok Moon, Prakash Basnet, Ik-Young Choi, Taeyoung Um, Doh Hoon Kim

**Affiliations:** 1College of Life Science and Natural Resources, Dong-A University, Busan 49315, Republic of Korea; wtyang@dau.ac.kr (W.T.Y.); busankd88@naver.com (K.D.B.); seonlee@dau.ac.kr (S.-W.L.); 2Graduate School of Green-Bio Science, Kyung Hee University, Yongin 17104, Republic of Korea; khjung2010@khu.ac.kr (K.H.J.); moonsun@khu.ac.kr (S.M.); 3Department of Agriculture and Life Industry, Kangwon National University, Chuncheon 24341, Republic of Korea; prakashbasnet2007@kangwon.ac.kr (P.B.); choii@kangwon.ac.kr (I.-Y.C.); 4Department of Agriculture and Life Science Institute, Kangwon National University, Chuncheon 24341, Republic of Korea

**Keywords:** MYB transcription factor, *Oryza sativa*, phosphorus homeostasis, Pi starvation response, salt stress tolerance

## Abstract

Inorganic phosphate (Pi) homeostasis plays an important role in plant growth and abiotic stress tolerance. Several MYB-CC transcription factors involved in Pi homeostasis have been identified in rice (*Oryza sativa*). PHOSPHATE STARVATION RESPONSE-LIKE 7 (PHL7) is a class II MYC-CC protein, in which the MYC-CC domain is located at the N terminus. In this study, we established that OsPHL7 is localized to the nucleus and that the encoding gene is induced by Pi deficiency. The Pi-responsive genes and Pi transporter genes are positively regulated by OsPHL7. The overexpression of *OsPHL7* enhanced the tolerance of rice plants to Pi starvation, whereas the RNA interference-based knockdown of this gene resulted in increased sensitivity to Pi deficiency. Transgenic rice plants overexpressing *OsPHL7* produced more roots than wild-type plants under both Pi-sufficient and Pi-deficient conditions and accumulated more Pi in the shoots and roots. In addition, the overexpression of *OsPHL7* enhanced rice tolerance to salt stress. Together, these results demonstrate that *OsPHL7* is involved in the maintenance of Pi homeostasis and enhances tolerance to Pi deficiency and salt stress in rice.

## 1. Introduction

Phosphorus (P), nitrogen (N) and potassium (K) are essential macro-elements vital for maintaining the growth, development and productivity of crops [1,2]. P is a major macronutrient that strongly influences plant growth and productivity, including in rice (*Oryza sativa*), a major crop worldwide [3,4]. Phosphorus stress negatively regulates metabolic processes [5,6,7], and plants have evolved various adaptive strategies that enable them to withstand this stress. To cope with limited phosphate availability, plants have developed a phosphate homeostasis mechanism that mobilizes and stores Pi from soil, which is utilized under stress [4]. Recent studies have reported that the interaction of N, P and K is important for nutrient homeostasis. Both N and P show a synergistic influence, wherein N has a positive impact on P uptake under normal conditions [8] but acts negatively during P starvation conditions [9,10]. In rice, there is an interaction between potassium (K) and phosphorus (P) that influences both grain yield and plant immunity [2,11]. Many plant genes are expressed under inorganic phosphate (Pi) deficiency, causing changes in various molecular, cellular and physiological processes [12,13]. As an example, plants alter their root structure in response to stress conditions, thus optimizing the surface area available for Pi absorption [14,15,16]. A recent report showed that the exogenous application of K represses Pi uptake and induces *PHOSPHORUS STARVATION RESPONSE* (*PSR*) genes [17]. 

Many *PSR* genes are reported as MYB transcription factors (TFs). MYB TFs are the largest TF family [18], and they consist of a conserved MYB domain at the N-terminus and are found in every eukaryote [19,20]. These TFs are also crucial regulators in response to Pi-deficient conditions in plants [21,22,23]. In rice, MYB TFs, such as PHOSPHORUS STARVATION RESPONSE 1 (OsPSR1), PHOSPHATE STARVATION RESPONSE 1 (OsPHR1), OsPHR2, OsPHR3, OsPHR4 and OsMYB2P-1, have been reported to regulate Pi starvation signaling and thus maintain Pi homeostasis. *Arabidopsis PHR1* and *OsPHR2* are involved in regulating the expression of PSR genes [23,24,25,26]. 

Plants that overexpress *OsPSR1* or *OsPHR1* exhibit increased Pi transport and accumulate more Pi compared to control plants under Pi-deficient conditions [23,26,27,28]. In *Arabidopsis thaliana*, *PHR-LIKE 1* (*AtPHL1*) regulates the balance of essential nutrients, such as sulfate [26], zinc [27] and iron (Fe) [28], for Pi homeostasis. The expression of *PHR1* and *OsPHR3* is activated by the addition of N supplementation [29,30,31,32]. In the *phr1 phl1* double-knockout mutant, the Pi starvation-responsive (PSI) genes are downregulated [31]. The overexpression of *OsPHR2* and *OsPHR3* improved root length, hair growth and the expression level of PSR genes . Also, OsPHL2 and OsPHL3 play roles in the regulation of Pi starvation signaling [23,32,33]. 

Recent studies have discovered several genes involved in Pi homeostasis, which constitutes progress toward understanding its regulation at the molecular level. It was found that Pi controls the expression level of microRNA [34]. The first microRNA found to be specifically triggered in response to Pi deficiency was miR399. Together, miR399 and its target gene, *PHOSPHATE 2* (*PHO2*), oppositely regulate Pi accumulation in plants [35,36]. The overexpression of *miR399* or the loss of *AtPHO2* lead to a high Pi accumulation in *Arabidopsis* [35,36,37]. *PHR1* induces the expression of *miR399* and represses the transcript level of *PHO2* and phosphate transporters (PTs) under nitrogen-limited conditions [38]. 

Salt stress is a major environmental limiting factor that affects plant growth and development [39] and results in nutritional and hormonal imbalance, ion toxicity, oxidative stress and increased plant susceptibility to diseases [40]. The homeostasis of Pi is closely involved in the defense mechanism against salt stress in plants [41,42,43]. For example, a higher accumulation of Pi and expression levels of various Pi transporters (13 Pi transport proteins (OsPT1–13)) in the *ospho2* mutant also increased salt stress tolerance in rice [44,45,46]. This enhanced tolerance to salinity was also observed in other plants with higher Pi contents [44]. Additionally, *OsMYB2* plays a role in enhancing salt tolerance in rice [45,47]. 

Many studies have shown that the class I TFs in the MYB-CC family (OsPHR1, OsPHR2, OsPHR3 and OsPHR4) are involved in Pi homeostasis in plants; however, the functions of the class II TFs in the response to Pi have not been identified and characterized. OsPHL7 is a class II TF in the MYB-CC family. Also, OsPHL7 was named PHOSPHATE STARVATION RESPONSE-LIKE 7 by its domain, but its function has not been studied. 

In this study, *OsPHL7*-overexpressing rice plants had increased Pi contents under both Pi-sufficient and Pi-deficient conditions, while the Pi concentrations in the *osphl7*-RNA interference (RNAi) knockdown mutants were lower than those in the wild-type (WT) plants. Furthermore, the expression of the PSR and Pi transporter genes, including *OsPT2*, *OsPT6*and *OsPT10*, was upregulated in the shoots and roots of the *OsPHL7-*overexpressing plants under both Pi-sufficient and Pi-deficient conditions. The *OsPHL7*-overexpressing plants also displayed an enhanced tolerance to salt stress, while the *ophl7*-RNAi mutants exhibited a reduced salt stress tolerance. Together with the findings that *OsPHL7* expression leads to the accumulation of Pi and increased salt stress tolerance in plants, our results indicate that Pi homeostasis is involved in the salt stress response in rice.

## 2. Results

### 2.1. OsPHL7 Expression Is Induced in Pi- and Fe-Deficient Conditions

To explore the function of *OsPHL7* in rice, we first analyzed the amino acid sequence of this protein using the Basic Local Alignment Search Tool (BLAST) and Conserved Domain Search Service tools from the National Center for Biotechnology Information (NCBI) (Appendix A). The N-terminal region of the OsPHL7 protein contains MYB-CC and coiled-coil domains. The 16 rice TFs containing MYB-CC domains were compared with OsPHL7. These TFs were grouped into three classes according to the location of the MYB-CC domain (Appendix A) [19,20]. The *OsPHL7* (Os06g0609500) gene was closely related to *OsMYCc* (Os09g0299200) in class II, which is involved in sodium (Na^+^) and potassium (K^+^) homeostasis in rice [45,47]. The class I genes, including *OsPHR1*, *OsPHR2*, *OsPHR3* and *OsPHR4*, play roles in Pi homeostasis in rice [48]. We determined the subcellular localization of OsPHL7 using a rice protoplast transient expression system, revealing that the fluorescent protein-tagged OsPHL7 localized to the nucleus (Appendix A). These results indicate that OsPHL7 is a TF potentially involved in the nutrient homeostasis system in rice.

To investigate the expression pattern of *OsPHL7* in rice, we analyzed transcription of whole plants in response to deficiencies of various mineral nutrients, including nitrogen (N), K, and Fe and Pi, using Northern blot assays (Figure 1A), revealing that *OsPHL7* was upregulated by Fe and Pi deficiency. The expression of *OsPHL7* was also analyzed in the roots and shoots under Pi deficiency at various time points (Figure 1B), revealing that it increased over time in both tissues, reaching a higher level in the root than in the shoot. These results indicate that the expression of *OsPHL7* is induced by a deficiency of Pi or Fe, suggesting that it plays a role in Pi and Fe homeostasis in rice.

### 2.2. OsPHL7 Is Involved in Pi Homeostasis in Rice

To investigate the function of *OsPHL7* in rice, we generated *OsPHL7-*overexpressing (Ox) and *osphl7*-RNA interference (RNAi) transgenic rice plants (Appendix A). The transgenic lines were identified using qRT-PCR analysis (Appendix A). *OsPHL7-*Ox lines 2, 3 and 5 (Ox2, Ox3 and Ox5, respectively) and *osphl7-*RNAi lines 3, 4 and 7 (i3, i4 and i7, respectively) were selected for further analysis. We analyzed the phenotypes and Pi contents of the *OsPHL7-*Ox plants, *osphl7-*RNAi plants and WT plants grown under Pi-sufficient (+Pi) and Pi-deficient (−Pi) conditions for 10 days (Figure 2A). The shoot length and fresh weight of the *OsPHL7-*Ox plants were higher than those of the WT plants under Pi-deficient conditions (Figure 2B,D). Similarly, the *OsPHL7-*Ox plants’ root fresh weight and root number were higher than those of the WT plants under Pi-deficient conditions, although their primary root lengths were not significantly different. In contrast, the shoot length, shoot fresh weight, root fresh weight and root number of the *osphl7-*RNAi plants were lower than those of the WT under Pi-deficient conditions (Figure 2B–E,H). These results suggest that *OsPHL7* enhances plant tolerance to Pi deficiency.

As many of the genes related to *OsPHL7* are involved in phosphate homeostasis, we hypothesized that the expression of *OsPHL7* would affect the phosphate concentration in plants. To investigate this, we analyzed the differences in the Pi contents of the *OsPHL7-*Ox and *osphl7-*RNAi plants under both Pi-sufficient and Pi-deficient conditions (Figure 2F,G). The Pi contents were dramatically higher in the shoots and roots of the *OsPHL7-*Ox plants than in the WT plants under both Pi-sufficient and Pi-deficient conditions. In contrast, the Pi contents in the shoots and roots of the *osphl7-*RNAi plants were similar to those of the WT plants under both Pi-sufficient and Pi-deficient conditions. These results together demonstrate that increased *OsPHL7* expression increases the accumulation of phosphate in plants.

### 2.3. OsPHL7 Regulates the Expression of PSR Genes

If *OsPHL7* is involved in Pi homeostasis during Pi starvation, we hypothesized that it might regulate the expression of several genes involved in the Pi starvation response, including Sulfoquinovosyldiacylglycerol 2 (*OsSQD2*), *OsPHR2*, induced by phosphate starvation 2 (*OsIPS2*), purple acid phosphatase 10 (*OsPAP10*), *OsmiR399a*, *OsmiR399j*, *OsPT2*, *OsPT6* and *OsPT10*. To address this, we analyzed the expression levels of these genes in the shoots (Figure 3) and roots (Appendix A) of the *OsPHL7*-Ox and *osphl7-*RNAi plants. All of these genes were upregulated in both the shoots and roots of the *OsPHL7-*Ox plants compared with the WT plants under both Pi-sufficient and Pi-deficient conditions. In contrast, these genes were downregulated in both the shoots and roots of the *osphl7-*RNAi plants in comparison with the WT plants. 

*OsPHO2* is targeted by *OsmiR399* in Pi starvation signaling [35,36,37], so we also analyzed the expression of this gene in the *OsPHL7* lines. In the shoots and roots of the *osphl7*-RNAi plants, *OsPHO2* was significantly upregulated compared with the WT plants under both Pi-sufficient and Pi-deficient conditions, while it was downregulated in the *OsPHL7-*Ox plants (Figure 3 and Appendix A). Together, these results indicate that *OsPHL7* regulates the expression of the PSI genes, suggesting that it is involved in Pi starvation signaling in rice. 

### 2.4. Overexpression of OsPHL7 Enhances Salt Stress Tolerance

Previous studies have reported that the accumulation of phosphate is involved in salt stress tolerance [46]. To examine whether *OsPHL7* plays a role in this phenomenon, we exposed 10-day-old *OsPHL7*-Ox, *osphl7-*RNAi and WT plants to salt stress (Figure 4A). After 10 days of salt stress, the *osphl7-*RNAi and WT plants were severely damaged, with most of their leaves curling and turning yellow. The *OsPHL7*-Ox plants were also damaged by the salt stress, but the visual symptoms were weaker than those of the WT plants. 

To further characterize the phenotypes, we analyzed the lengths, fresh weights and survival rates of the different plant lines grown under salt stress. The lengths and fresh weights of the *OsPHL7*-Ox shoots and roots were greater than those of the WT plants (Figure 4B,C). The survival rate of the *OsPHL7*-Ox plants was also higher than that of the WT plants under salt stress (Figure 4D), indicating that the overexpression of *OsPHL7* enhances salt stress tolerance. In contrast, the lengths and fresh weights of the *osphl7-*RNAi roots and shoots were similar to those of the WT plants (Figure 4B,C); however, the survival rate of the *osphl7-*RNAi plants was significantly reduced under salt stress (Figure 4D), indicating that the *osphl7-*RNAi mutant plants are more susceptible to salt stress. Collectively, these findings suggest that *OsPHL7* expression promotes salt stress tolerance in rice.

## 3. Discussion

An understanding of the phosphorus (P), nitrogen (N) and potassium (K) interactions is important to overcome the constraints to efficient crop yields [1,10]. The presence of nitrogen (N) actively regulates the genes involved in the response to phosphate starvation in plants [32,49]. Recent studies have shown antagonistic interactions between P, N and K [1,49,50,51,52,53]. During Pi-deficit conditions, N supplementation activates PSR, while it strongly inhibits PSR under N starvation conditions [52,53]. A study of N and phosphate (P) crosstalk showed that low N availability conditions result in an upregulation of phosphate-responsive genes (PSR) and Pi concentration in maize [53]. The expression of many PSR genes is antagonistically regulated by N- and Pi-deficient conditions [53], and *PHR1* and *OsPHR3* transcripts are significantly reduced in N-deficient conditions [29,30,31,32]. Our results showed that the expression of *OsPHL7* was downregulated in N-deficient conditions and increased in Pi-deficient conditions (Figure 1A). A report showed that under high K conditions, Pi uptake is inhibited, which induces the expression of PSR genes [52]. The expression of phosphate transporters (PTs) showed upregulation under Pi-deficient conditions and downregulation under K-deficient conditions in tomato plants [17]. Also, we demonstrated that the expression of *OsPHL7* was reduced in K-deficient conditions (Figure 1A). These findings suggest that *OsPHL7* is involved in regulating phosphorus (P), nitrogen (N) and potassium (K) homeostasis.

Previous studies have shown that the class I TFs in the MYB-CC family (OsPHR1, OsPHR2, OsPHR3 and OsPHR4) are involved in Pi homeostasis in plants; however, the functions of the class II and III TFs in the response to Pi have not been identified or characterized [6,54]. In this study, we identified a novel rice gene, *OsPHL7*, encoding a class II MYB-CC family protein. We characterized the phenotypes of the *OsPHL7-*Ox and *osphl7-*RNAi plants, which revealed that *OsPHL7* regulates the responses to Pi deficiency and salt stress. Furthermore, we found that the expression of *OsPHL7* is strongly upregulated in response to Pi deficiency (Figure 1B). These findings suggest that *OsPHL7* is involved in Pi homeostasis in rice, which was partially supported by our observation of the positive correlation between the expression patterns of the Pi transporter genes and *OsPHL7* (Figure 3 and Appendix A). The Pi transport genes play crucial roles in the accumulation of Pi by regulating a plant’s response to Pi deficiency [6,54]. Our results therefore demonstrate that *OsPHL7* is involved in regulating the Pi transporters that mediate Pi homeostasis in rice. 

A previous study showed that Fe and Pi are interdependent in regulating photosynthesis [55]. In Pi starvation conditions, plants reduce the expression of the genes in response to Fe deficiency in *Arabidopsis* [56]. The joint application of Fe and Pi significantly increased the yield of tomato plants [57]. A report showed that *PHR1* and *PHL1* interact for Fe and Pi nutrient signals, suggesting that both genes are involved in Pi and Fe homeostasis [28]. The results showed that the expression of *OsPHL7* is upregulated in both Pi- and Fe-deficient conditions (Figure 1A), and the *OsPHL7-*Ox plants showed that the chlorophyll contents of rice plants improve under Fe-deficient conditions, whereas the *osphl7-*RNAi plants displayed yellow leaves and reduced chlorophyll contents under this stress (Appendix A). These results suggest that *OsPHL7* is important for mediating Pi and Fe homeostasis and chlorophyll formation in rice under Fe-deficient conditions. 

In rice, Pi starvation signaling is regulated by the expression of PSR genes (such as *OsSQD2*, *OsPHR2*, *OsIPS2*, *OsPAP10*, *OsPHO2*, *OsmiR399a*, *OSmiR399j*, *OsPT2*, *OsPT6* and *OsPT10*), leading to controlled Pi-deficient signaling and Pi uptake [23,24,25,38,58,59]. Under Pi-sufficient conditions, the expression of *PHRs* was upregulated in the shoots and roots of the *OsPHL7*-Ox plants, while the expression of *PHRs* was reduced in the shoots and roots of the *osphl7-*RNAi plants. Furthermore, in the *OsPHL7*-Ox plants, the expression of PHRs was induced under Pi-deficient conditions more than under normal conditions (Figure 3 and Appendix A). These findings suggest that OsPHL7 plays a role in Pi-deficient signaling in rice.

Recently, Na^+^-coupled Pi transporters (NaPi) have been identified in algae and plants [60,61,62]. NaPi plays a role in mediating Pi homeostasis in mammals [63]. And phosphate accumulation in plants increases salt stress tolerance. The *siz1* and *pho2* mutants, which cause more accumulation of Pi than in a WT plant, reduce Na^+^ uptake and Na^+^ contents in leaves [46]. Also, the exogenous supplementation of Pi improves salt tolerance in maize [64]. We also showed that the overexpression of *OsPHL7* is sufficient to improve salt stress tolerance (Figure 4). The accumulation of Pi is closely correlated with Na^+^ accumulation in plants, with increased Pi levels enhancing salt stress tolerance [46,59,60,61,62]. Here, we showed that the overexpression of *OsPHL7* increased the Pi content of rice (Figure 2), as well as the expression of *OsmiR399a* and *OsmiR399j,* while *OsPHO2*, targeted by *OsmiR399*, was downregulated (Figure 3 and Appendix A). Here, the Ox lines showed an upregulation of PT genes (OsPT2, OsPT6 and OsPT10), which might confer improved salt tolerance in plants. An elevation of Pi helped modulate nutrient uptake, contributing to enhanced tolerance to salt stress [65]. These observations suggest that *OsPHL7* might be involved in Na and NaPi homeostasis during salt stress. 

It is largely unknown how TFs regulate Pi, Fe and Na homeostasis in rice under nutrient-deficient conditions; however, the function of *OsPHL7* suggests that the regulation of cellular osmolality or ion homeostasis might be involved in the *OsPHL7*-mediated improvement of abiotic stress. Further molecular approaches will expand our understanding of the mechanisms underlying this process. 

## 4. Materials and Methods

### 4.1. Phylogenetic Analysis

The homologous sequences of *OsPHL7* (*Os06g0609500*) were obtained from the National Center for Biotechnology Information BLAST (https://blast.ncbi.nlm.nih.gov/Blast.cgi (accessed on 17 March 2018)). Based on amino acid sequences, multiple protein sequence alignment and phylogenetic trees were performed using the Clustal Omega program (https://www.ebi.ac.uk/Tools/msa/clustalo/ (accessed on 17 March 2018)) and MEGA X (https://www.megasoftware.net/ (accessed on 17 March 2018)).

### 4.2. Generation of OsPHL7 Transgenic Plants and Growth Conditions

To generate overexpressing and RNA interference transgenic rice, full-length cDNA (990 bp) and a partial fragment (298 bp) of the *OsPHL7* gene were inserted into *pENTR™/D-TOPO* (Invitrogen, Carlsbad, CA, USA), respectively. The recombination reaction was carried out using a Gateway Cloning system (Invitrogen, Carlsbad, CA, USA). The destination vectors used were a *pH7WG2D.1* vector containing the hygromycin resistance gene and a *pB7GWIWG2.0* vector containing the *Bar* resistance gene. The *OsPHL7:pH7WG2D.1* (*OsPHL7*-Ox) and *OsPHL7*(sense)- *OsPHL7*(anti-sense)*:pB7GWIWG2.0* (*OsPHL7*-RNAi) constructs were introduced into *Agrobacterium tumefaciens* (EHA105) by electroporation, then transformed into rice. 

*Oryza sativa* L. ‘Dongjin’ was used in all physiological experiments (WT control) and to generate the transgenic plants. Transgenic rice was selected for the T3 generation and used in all experiments. We used a modified version of a general rice transformation protocol, and hydroponic culture experiments were performed . Ten-day-old seedlings of wild-type, *OsPHL7-*OX and *osphl7-*RNAi transgenic rice were submerged in an MS medium supplemented with 100 mM of NaCl for 10 days. Phenotypes, lengths, weights, survival rates and other traits were then examined during the salt treatment period. Rice was grown in a growth chamber at 32 °C with 6 h/8 h (light/dark) light conditions on MS (Murashige and Skoog) medium for 7 days. Subsequently, seedlings were cultured in Hoagland solution containing sufficient nutrients (containing 500 μM of KH_2_PO_4_, 5 mM of Ca(NO_3_)_2_∙4H_2_O, 2.5 mM of K_2_SO_4_, 1 mM of Fe-EDTA) and solutions deficient in each nutrient (20 μM of KH_2_PO_4_ (−Pi), 250 μM of Ca(NO_3_)_2_∙4H_2_O (−N), 10 µM of K_2_SO_4_ (−K), 10 µM of Fe-EDTA (−Fe)). 

### 4.3. Pi Content Measurement

The shoot and root fresh weights were measured, after which the plants were dried at 80 °C for 3 days. The Pi content was measured as described previously [23]. A dried 100 mg sample was homogenized in 1 mL of 10% (*w*/*v*) perchloric acid (PCA) using an ice-cold mortar and pestle. The homogenized samples were then diluted 10-fold with 5% (*w*/*v*) PCA and placed on ice for 30 min. After centrifugation at 10,000× *g* for 10 min at 4 °C, the Pi content of the supernatant was measured using the molybdate blue method, for which 0.4% (*w*/*v*) ammonium molybdate melted in 0.5 M of H_2_SO_4_ (solution A) was mixed with 10% ascorbic acid (solution B) (A:B = 6:1). A 2 mL aliquot of this solution was added to 1 mL of the sample solution and incubated in a water bath at 40 °C for 20 min. After cooling on ice, the absorbance was measured at 820 nm, using KH_2_PO_4_ for the standard curve.

### 4.4. RNA Extraction and Northern Blot Assay

Total RNA was extracted from 100 mg of shoots and roots using TRIzol reagent (MilliporeSigma, Burlington, MA, USA), according to the manufacturer’s instructions, for a Northern blot analysis. A 20 µM aliquot of total RNA was subjected to electrophoresis and separated on a 1.2% (*w*/*v*) denaturing formaldehyde agarose gel, then transferred onto a Hybrid-N^+^ membrane (GE Healthcare, Chicago, IL, USA) and cross-linked using a commercial UVP UV-light crosslinking instrument (Analytik Jena, Upland, CA, USA). The membrane was hybridized overnight at 65 °C with a [^32^P]-dCTP-labeled probe (Stratagene; Agilent Technologies, Santa Clara, CA, USA) in a solution containing 20% (*w*/*v*) sodium dodecyl sulfate (SDS), 20× SSPE, 100 g/L of polyethylene glycol (PEG; 8000 mwt), 250 mg/L of heparin and 10 mL/L of herring sperm DNA. Each probe for the *OsPT8* gene was generated using a primer set targeting the open reading frame. The membranes were washed twice in 2× saline–sodium citrate (SSC) and 0.2% (*w*/*v*) SDS at 65 °C for 10 min, twice with 1× SSC and 0.2% (*w*/*v*) SDS at 65 °C for 10 min and once in 0.1× SSC and 0.2% (*w*/*v*) SDS at 65 °C for 20 min. The dried membranes were placed on X-ray films, which were exposed at −72 °C for 1 day before being developed.

### 4.5. Reverse Transcription Quantitative Polymerase Chain Reaction (RT-qPCR) 

Total RNA was extracted with an RNeasy Kit (Qiagen, Hilden, Germany) according to the manufacturer’s instructions for RT-qPCR analysis . First-strand cDNAs were synthesized using 1 µg of total RNA with a cDNA Synthesis Kit (Takara, Kusatsu, Shiga, Japan) to serve as the templates for RT-qPCR. To analyze the gene expression levels, RT-qPCR was performed, and values were automatically calculated using a CFX94 Real-time PCR Detection System and CFX Manager software (Bio-Rad, Hercules, CA, USA) following a standard protocol. *OsActin1* was used as a reference gene for RT-qPCR. Three technical replicates of the RT-qPCRs were performed using three biological replicates. The sequences of primers used in the RT-qPCR analysis are provided in Appendix A. 

### 4.6. Chlorophyll Content Measurement

To measure the chlorophyll content, 300 mg of leaf samples were ground and incubated in 5 mL of 80% acetone in the dark for 30 min. After being centrifuged at 4 °C for 15 min at 3000 rpm, 200 µL of the supernatant was transferred onto a 96-well microplate. The total chlorophyll content was measured using a SPECTROstarNano (BMG Labtech, Ortenberg, Germany), and the absorbance was measured at 645 and 663 nm. The chlorophyll concentrations were calculated as described previously [66].

### 4.7. Cellular Localization of OsPHL7 

The recombinant DNA (pHBT-GFP-OsPHL7) was generated by introducing amplified cDNAs into NotI/PstI-digested plasmids using an In-Fusion^®^ HD Cloning Kit (Takara). The protoplasts were extracted from seedlings (10-day-old rice), grown in 1MS (Murashige and Skoog) in a growth chamber in the dark at 28 °C, and then transformed with pHBT-GFP-OsPHL7, as described previously [67], with a slight modification. Around 2.5 × 10^6^ protoplasts were transformed with 1 μg of pHBT-GFP-OsPHL7. The transformed protoplasts were incubated in the dark at 28 °C for 10 h and analyzed using an SP8 STED laser scanning confocal microscope (Leica Microsystems, Mannheim, Germany). The GFP was detected between 470 and 550 nm.

## Figures and Tables

**Figure 1 plants-13-00637-f001:**
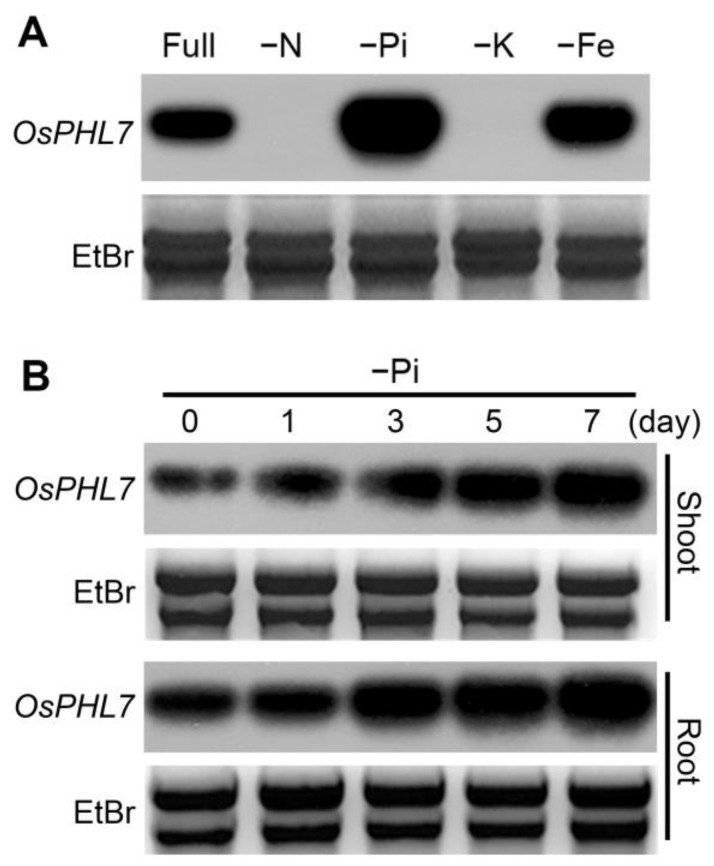
*OsPHL7* expression patterns in response to N-, K-, Fe- or Pi-deficient conditions. (**A**,**B**) Northern blot assays of the expression patterns of *OsPHL7* in whole plants (**A**) or in the shoots and roots (**B**) of ten-day-old wild-type (WT) plants. (**A**) Expression levels in response to N, K, Fe or Pi deficiency after 1 day. (**B**) Time course of *OsPHL7* expression in response to Pi deficiency. “Full” indicates the nutrient-sufficient conditions, which included 5 mM of N, 2.5 mM of K, 1 mM of Fe and 500 µM of Pi. For each nutrient-deficient condition, one of the nutrients was reduced to 250 µM of N, 10 µM of K, 10 µM of Fe, or 20 µM of Pi, with full levels of the other nutrients provided. The treatments were applied to 10-day-old WT plants. Ethidium bromide (EtBr) staining shows equal loading of RNA.

**Figure 2 plants-13-00637-f002:**
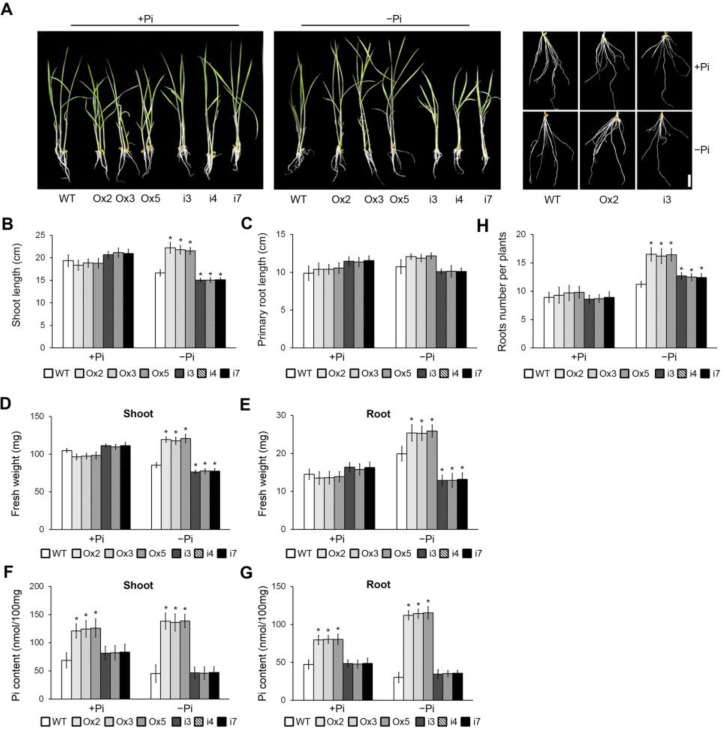
Effect of *OsPHL7* on the phenotypic responses to Pi deficiency. (**A**) The wild-type (WT), *OsPHL7*-overexpressing (Ox) and *osphl7*-RNA interference (RNAi) plants were grown for 10 days, after which seedlings were transferred to Pi-sufficient and Pi-deficient conditions. Scale bars indicate 5 cm (left, mid) and 1 cm (right). (**B**–**E**) Quantification of the lengths of the shoots (**B**) and primary roots (**C**) and the fresh weights of the shoots (**D**) and roots (**E**). (**F**,**G**) Pi concentrations were measured in the shoots (**F**) and roots (**G**) of plants under both Pi-sufficient and Pi-deficient conditions. (**H**) Quantification of root numbers in *OsPHL7*-Ox and *osphl7*-RNAi plants. Data represent mean values ± SD (*n* = 30). Asterisks indicate statistically significant differences between the corresponding samples and their controls (*p* < 0.01, 1-way ANOVA with Tukey post hoc test).

**Figure 3 plants-13-00637-f003:**
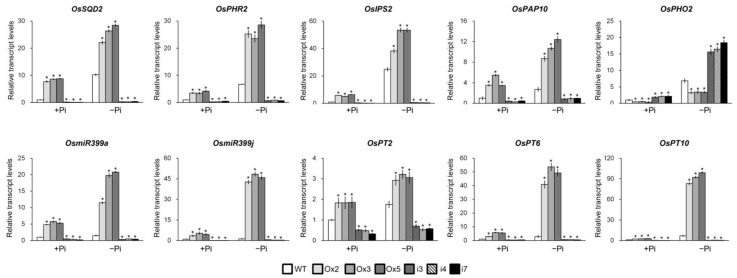
The expression patterns of Pi starvation response genes and Pi transport genes in the shoot of *OsPHL7*-overexpressing (Ox) and *osphl7*-RNA interference (RNAi) plants under Pi sufficient or deficiency conditions. RT-qPCR analysis of the expression levels of *OsSQD2*, *OsPHR2*, *OsIPS2*, *OsPAP10*, *OsPHO2*, *OsmiR399a*, *OSmiR399j*, *OsPT2*, *OsPT6* and *OsPT10* in response to Pi-deficient conditions in the shoots of 10-day-old *OsPHL7*-overexpressing (Ox), *osphl7*-RNA interference (RNAi) and wild-type (WT) plants. The plants were treated with 500 µM of Pi (+Pi) or 20 µM of Pi (−Pi) for 1 day. Ox2, Ox3 and Ox5 are three independent lines of P35S::*OsPHL7*; i3, i4 and i7 are three independent lines of *osphl7*-RNAi. The data are mean values of three biological replicates, and error bars indicate SD. Asterisks indicate statistically significant differences between the corresponding samples and their controls (*p* < 0.01, 1-way ANOVA with Tukey post hoc test). *OsActin1* was used as the internal control, and the relative expression levels are shown in fold values.

**Figure 4 plants-13-00637-f004:**
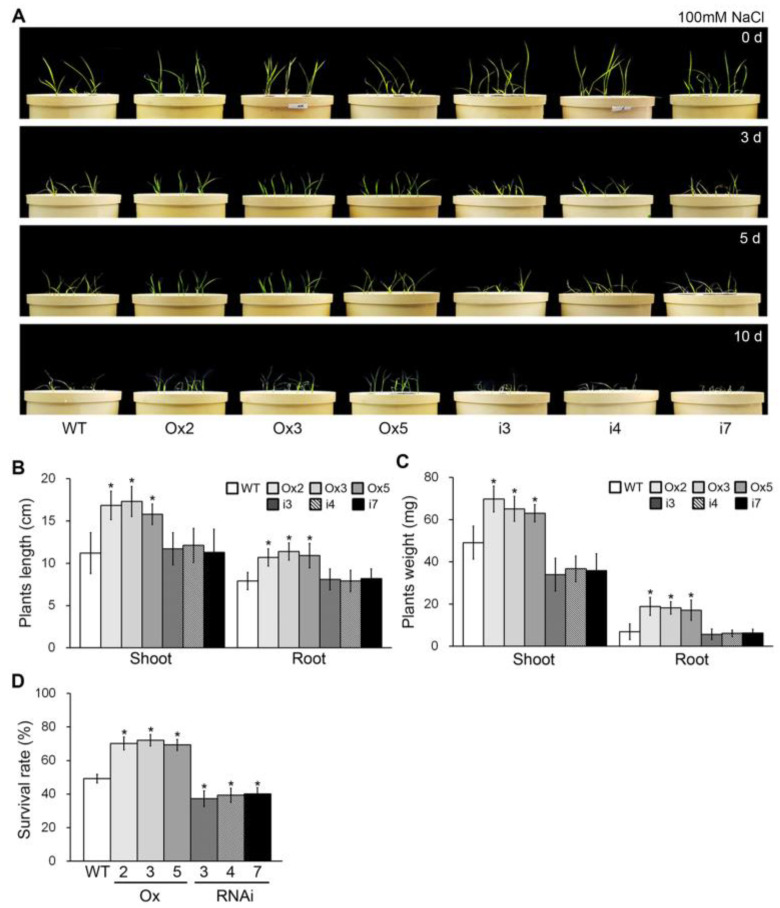
Effect of *OsPHL7* on salt stress tolerance. (**A**) The phenotypes of the *OsPHL7-*overexpressing (Ox), *osphl7*-RNA interference (RNAi) and wild-type (WT) plants during salt stress. Ten-day-old plants were exposed to salt stress (100 µM NaCl) for a further 10 days. (**B**–**D**) Measurement of the lengths (**B**), weights (**C**) and survival rates (**D**) of the plants after 10 days of salt stress. The data represent mean values ± SD (*n* = 30). Asterisks indicate statistically significant differences between the corresponding samples and their controls (*p* < 0.01, Student’s *t*-test).

## Data Availability

Data are contained within the article and Appendix A.

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
