# Peer review of "The MYB-CC Transcription Factor PHOSPHATE STARVATION RESPONSE-LIKE 7 (PHL7) Functions in Phosphate Homeostasis and Affects Salt Stress Tolerance in Rice"

_plants, 2024, doi:10.3390/plants13050637_

Round 1

Reviewer 1 Report (Previous Reviewer 1)

Comments and Suggestions for Authors

This submission is based on the previous version, and the author has made some modifications. After reviewing the revised manuscript, it can be seen that the author's revisions were not enough for scientific publication. It is necessary to make further improvements and supplements on the discussion section, because it was too brief and did not delve into in-depth scientific issues in current version. Authours should cite some previous relevant investigations and discuss them.

Comments on the Quality of English Language

Further improvements are still needed in the English language.

Author Response

Reviewer 2 Report (Previous Reviewer 2)

Comments and Suggestions for Authors

The authors have made corespondent modifications. I don’t have any other questions.

Author Response

Reviewer 3 Report (Previous Reviewer 3)

Comments and Suggestions for Authors

In the revised version of the manuscript entitled “The MYB-CC transcription factor PHOSPHATE STARVATION 2 RESPONSE-LIKE 7 (PHL7) functions in phosphate homeostasis 3 and affects salt stress tolerance in rice”, the authors addressed many, but not all my concerns.

Remaining concerns:

When I asked to explain why authors selected specifically OsPHL7 for this study, I meant to make this clearer in the Introduction/Results part, rather than explaining this to me.

But my main concerns are the Northern blot and the NaCl treatment images.

I am concerned about the Northern Blot showing a complete lack of expression after just one day in -N and -K. While authors state in their address to me that they have repeated this experiment more than three times and imply that this result is reproducible, no additional Northern blots are shown or even mentioned in the text. With a result as unusual as this is, I believe it would be warranted to give additional support by showing INDEPENDENT biological replications in the supplemental material.

Another point of concern is still the NaCl treatment time course experiment. We see different images than in the last manuscript, but I’m puzzled by the plant sizes; do plants substantially shrink in the first 3 days of NaCl treatment? The 10 day image looks quite intriguing, with overexpression lines looking slightly healthier, and RNAi plants slightly worse than the wildtype, but I don’t understand the change from 0 to 3 days.

Round 2

Reviewer 1 Report (Previous Reviewer 1)

Comments and Suggestions for Authors

Review comments:

Compared to the previous version, the author has indeed made significant improvements to the manuscript, e.g. introduction and discussion sections. However, upon closer reading of the revised version, it was found that there are still many flaws and inconsistencies in the manuscript. Based on the above reasons, the manuscript still needs to be improved thoroughly.

1.I couldn't find any relevant files in the author's submission file, as indicated by author in L406-411: Figure S1; Figure S2; Figure S3; Figure S4.  The author should submit relevant data for evaluation.

2. L408-410: Figure S3: In root, effect of OsPHL7 on the expression of the Pi starvation–responsive genes and Pi transport genes under Pi-sufficient and Pi-deficient conditions. It is difficult to understand the author's exact meaning from this caption.

3. Considering that the supporting data in the main text of the manuscript is a bit thin and inadequate, the author should consider placing some of the supplementary data that are closely related to the theme as the main figure of the manuscript, eg. Figure S1 and Figure S3. But this is just a small suggestion, the author can make their own decision.

4. In figure 4, the author only presented data for one OE line (ox2) and one RNAi line (i3), which lacks credibility and is very imprecise. This may be the biggest flaw currently present in the manuscript. Does it mean that other lines such as OX3 and OX5 do not possess such characteristics? To ensure the credibility and consistency of the experimental results, the author needs to supplement the salt stress resistance data of OX3 and OX5 and RNAi lines. Without these data, the manuscript deserves rejection.

Comments on the Quality of English Language

The language quality is relatively poor.

Author Response

Reviewer 3 Report (Previous Reviewer 3)

Comments and Suggestions for Authors

Authors addressed my concerns sufficiently; I believe the current version is acceptable for publication.

Author Response

Authors addressed my concerns sufficiently; I believe the current version is acceptable for publication.

>Response

Thank you for your comment.

Round 3

Reviewer 1 Report (Previous Reviewer 1)

Comments and Suggestions for Authors

In the revised version, the author has made further improvements to their manuscript. My concerns have basically improved, although there are still some minor flaws in certain parts of the manuscript. For example,the statistical testing method used by the author throughout the entire manuscript is student's t-test. However, it is not appropriate in comparing more than two sets of samples using student's t-test.  The authors should use other statistical methods such as analysis of variance, rather than T-tests. After revising this issue, I think the manuscript can be reluctantly accepted for publication.

Author Response

This manuscript is a resubmission of an earlier submission. The following is a list of the peer review reports and author responses from that submission.

Round 1

Reviewer 1 Report

Comments and Suggestions for Authors

The manuscript entitled " The MYB-CC transcription factor PHOSPHATE STARVATION RESPONSE-LIKE 7 (PHL7) functions in phosphate homeostasis and affects salt stress tolerance in rice” is within the aim and scope of Plants.

The overall content of the article is relatively coherent and can basically explain the theme. Here are a few suggestions.

1. In result 2.1, the protein structure, evolutionary relationship, and subcellular localization of OsPHL7 were analyzed, and it was also found that OsPHL7 expression was upregulated in Fe and Pi deficiency. Therefore, the title of this paragraph only states that OsPHL7 expression is inappropriate in Pi deficiency.

2. What is its phenotype after treatment at other stages, as only the seedling stage was treated in the text? For example, the salt resistance during the tillering stage. Alternatively, the author can provide specific periods of response of the gene to Pi deficiency and salt stress.

3. Is the keyword MYB transcription factor or MYB-CC transcription factor?

4. The following experimental methods need to be supplemented: constructing evolutionary trees, subcellular localization, and salt stress treatment.

5. Writing standards should be clear, including the use of regular and italics, eg: Oryza sativa in line 22 and the name of interfering plants, as well as the citation format for references in line 53 of the article. Please carefully check the formatting issue of the entire article.

6. The first paragraph in the discussion section is not related to the topic of the article. Please carefully check the content of this paragraph and delete it.

Reviewer 2 Report

Comments and Suggestions for Authors

This article provides a preliminary verification of the functionality of OsPHL7, but there are still shortcomings.

1.Why do wild types with low Pi grow better than those with high Pi in Figure 2

2.Is the planting mode in Figure 4 single row or double row? Why is the wild type single row while the other two groups are double row. And the plants before and after treatment are inconsistent.

Reviewer 3 Report

Comments and Suggestions for Authors

The manuscript entitled “The MYB-CC transcription factor PHOSPHATE STARVATION 2 RESPONSE-LIKE 7 (PHL7) functions in phosphate homeostasis 3 and affects salt stress tolerance in rice” by Yang et al. explores the possible role of OsPHL7 in phosphate deficiency and salt tolerance in rice. My main concern in regard to this paper is the underlying data; frankly, many of the results presented in the various figures do not convince me. For example, a complete lack of expression of OsPHL7 after one day under -N and -K , while it does show background expression in complete medium, is making me wonder if something went wrong. Or a reported increase in chlorophyll content on OsPHL7-overexpression plants, while the associated photo shows rather yellow plants, compared to the wild type. For many experiments, results of only 1 overexpression line and 1 knockdown-line are reported, even though 3 assumingly independent transformations have been selected. In the qRT-PCR experiment, where results of all 3 lines are shown, the lack of variability between lines makes me wonder if these are truly independently transformed lines. Unless my concern regarding the data has been sufficiently addressed, I don’t think this manuscript is ready for publication.

I have outlined my detailed questions and concerns below.

INTRODUCTION

In Arabidopsis thaliana, PHR-50 LIKE 1 (AtPHL1) regulates the balance of essential nutrients, such as sulfate [17], zinc [18], and iron (Fe) [19], using Pi homeostasis.

What does “using Pi homeostasis” mean here? Clarify.

RESULTS

Explain why you selected specifically OsPHL7 for this study.

“These TFs were grouped into three classes according to the location of the MYB-CC do-85 main (Supplementary Figure S1A).”

Was this done previously? In that case a citation should be added. 

Or did you do this grouping, in which case more details on location of domains in the 3 classes should be added.

“We determined the subcellular localization of OsPHL7 using a rice protoplast transient expression system, revealing that the fluorescent protein–tagged OsPHL7 localized to the nucleus.”

More details should be given; I didn’t find the procedure described in the methods section. Or was this done previously? In which case that should be made clear and appropriate citation(s) should be added.

Fig. 1A is strange; no expression under -N and -K? There is some expression in the control (“full”); any explanation why expression completely disappears after just one day under -N and -K? Could this be an artifact/mistake? Was the Northern blot done in replicates?

Fig 2B

Why only showing the results of one of the three transgenic overexpression lines? Did the other two lines not show any differences? This seems like cherry-picking; if one has 3 independent lines, the results of all 3 lines should be presented. The same is true for the RNAi lines.

Fig. 3

Well, the results among the independently transformed lines look so similar, more what I am used to from technical replicates. Are the authors sure that these are the results from independently transformed lines? Any explanation why there is not the usual variability of fold-change expression data?

Apparently, only one gene (OsActin1) was used for normalization. Have authors made sure that this reference gene shows stable expression among the various transgenic lines?

“To examine whether OsPHL7 plays a role in this phenomenon, we exposed 10-day-old OsPHL7-Ox, osphl7-RNAi, and WT plants to salt stress (Figure 4A).” 

How exactly was salt stress imposed, and at what concentration? I also looked in the methods section for this information but couldn’t find it.

“The chlorophyll contents and survival rate of the OsPHL7-Ox plants were also higher than those of the WT plants under salt stress (Figure 4D and E),”

The “higher chlorophyll content” of OsPHL7-Ox plants is not supported by the photo, in which salt-stressed OsPHL7-Ox plants look more yellow than the wild type. 

Why do plants and pots look different between the time points? I assumed the photos would show the same plants at different time points, but judging from the looks of the plants and the color of the pots, that doesn’t seem to fit; why not?

DISCUSSION

Take out the first paragraph of instructions “Authors should discuss the…”

“Previous studies showed that the class I TFs in the MYB-CC family (OsPHR1, 250 OsPHR2, OsPHR3, and OsPHR4) are involved in Pi homeostasis in plants…”

Add appropriate citation’s)

“This suggests that OsPHL7 influences photosynthesis by mediating Fe and Pi homeostasis.” This seems a big jump to a conclusion.

MATERIALS AND METHODS

Add more detail. How was salt stress imposed? What was used as reference gene for qRT-PCR? How was the subcellular localization experiment done?

Comments on the Quality of English Language

I was sometimes unclear if presented results were from this study or from previous studies; adding some text and if appropriate citations should clarify this.